# Future Perspectives of Newborn Screening for Inborn Errors of Immunity

**DOI:** 10.3390/ijns7040074

**Published:** 2021-11-02

**Authors:** Maartje Blom, Robbert G. M. Bredius, Mirjam van der Burg

**Affiliations:** 1Laboratory for Pediatric Immunology, Department of Pediatrics, Willem-Alexander Children’s Hospital, Leiden University Medical Center, 2300 RC Leiden, The Netherlands; m.van_der_burg@lumc.nl; 2Department of Pediatrics, Willem-Alexander Children’s Hospital, Leiden University Medical Center, 2300 RC Leiden, The Netherlands; r.g.m.bredius@lumc.nl

**Keywords:** newborn screening, inborn errors of immunity, severe combined immunodeficiency, TREC, KREC, epigenetic immune cell counting, next-generation sequencing

## Abstract

Newborn screening (NBS) programs continue to expand due to innovations in both test methods and treatment options. Since the introduction of the T-cell receptor excision circle (TREC) assay 15 years ago, many countries have adopted screening for severe combined immunodeficiency (SCID) in their NBS program. SCID became the first inborn error of immunity (IEI) in population-based screening and at the same time the TREC assay became the first high-throughput DNA-based test in NBS laboratories. In addition to SCID, there are many other IEI that could benefit from early diagnosis and intervention by preventing severe infections, immune dysregulation, and autoimmunity, if a suitable NBS test was available. Advances in technologies such as KREC analysis, epigenetic immune cell counting, protein profiling, and genomic techniques such as next-generation sequencing (NGS) and whole-genome sequencing (WGS) could allow early detection of various IEI shortly after birth. In the next years, the role of these technical advances as well as ethical, social, and legal implications, logistics and cost will have to be carefully examined before different IEI can be considered as suitable candidates for inclusion in NBS programs.

## 1. Introduction

Expansion of newborn screening (NBS) with new disorders is driven by the development of new test modalities and treatment options. One of the more recent developments was the introduction of the first high-throughput DNA-based NBS test in the screening laboratory for the detection of severe combined immunodeficiency (SCID). SCID is one of the most severe forms of inborn errors of immunity (IEI) characterized by the absence or dysfunction of T-lymphocytes affecting both cellular and humoral immunity [1,2]. SCID or Combined immunodeficiency (CID), which is generally less profound than SCID, is a term used to describe a variety of genetic defects in more than 50 genes [3,4]. Infants with SCID typically appear normal at birth but develop severe infections in the first months of life. Without curative treatment, in the form of allogeneic hematopoietic stem cell transplantation (HSCT) or in some specific forms of SCID, gene therapy, affected infants die within the first year of life. Clearly, early definitive treatment, before the onset of infections, has the best outcome [5,6]. Due to the severity of the disease, an asymptomatic status early in life and improved survival and outcome after an early diagnosis, SCID was considered a suitable candidate for NBS. NBS for SCID is based on the detection of T-cell receptor excision circles (TRECs) with (q)PCR. TRECs are formed as a byproduct in approximately 70% of developing αβ T-lymphocytes and can therefore serve as a marker for thymic output [7,8]. Since the introduction of the TREC assay 15 years ago, many countries have introduced SCID in their NBS programs leading to improved outcomes for SCID patients worldwide [6,9].

SCID became the first immune disorder in the NBS program. However, in addition to SCID, there are many other IEI that could benefit from early diagnosis and intervention if a suitable NBS test was available. IEI are a heterogenous group of disorders characterized by an increased susceptibility to severe and/or recurrent infections, due to genetic defects affecting the development and/or function of the immune system [10]. Autoimmunity, autoinflammation, allergy, and malignancy can be common, and in some cases, predominant, clinical manifestations [11]. More than 430 IEI have been described with the discovery of new IEI occurring at an impressive rate [12]. With the Wilson and Jungner screening criteria in mind, several IEI would qualify as serious conditions that cause an important health problem and would benefit from early detection and treatment by preventing severe infections, immune dysregulation, and auto-immunity [13,14]. For some monogenetic IEI allogeneic HSCT might be a curative approach and autologous gene therapy could serve as a possible alternative treatment in the future [15]. This review will present future perspectives and recent technological advances that can potentially lead to expanded NBS for IEI.

## 2. Newborn Screening for (X-Linked) Agammaglobulinemia

One of the first steps in the expansion of NBS programs for IEI would be the implementation of NBS for X-linked agammaglobulinemia (XLA) and autosomal recessive XLA-like disorders. Agammaglobulinemia refers to a group of IEI in which B-cells are absent or dysfunctional, resulting in severely decreased or absent levels of all classes of serum immunoglobulins (Igs) and an inability to produce specific antibodies [16]. The most common form of this disease is XLA, caused by mutations in the Bruton’s tyrosine kinase (BTK) gene. BTK is a signal-transducing protein, thus mutations in the BTK gene cause a block in the differentiation of B-cell progenitors into mature B-cells affecting humoral immunity [17,18,19]. Patients with agammaglobulinemia develop serious recurrent infections from the sixth month of life, predominantly in the respiratory tract and the gastrointestinal tract [20,21]. Moreover, patients are at risk for severe meningoencephalitis caused by enteroviruses or life-threatening sepsis [22,23,24]. Without treatment, agammaglobulinemia can lead to chronic lung disease and permanent lung damage, such as bronchiectasis, and even premature mortality due to severe infections and complications [22,25,26]. Treatment consists of life-long administration of Igs either intravenously or subcutaneously combined with prophylactic antibiotics if indicated [27]. Early detection of these severe B-cell deficiencies and timely initiation of Ig replacement therapy is crucial to prevent secondary complications, long-term morbidity, and consequently mortality [25,28]. Previous studies have shown an increased incidence of chronic lung disease in patients with delayed diagnosis with a significant impact on prognosis and quality of life suggesting that NBS for XLA and other B-cell deficiencies would almost certainly result in improved clinical outcomes and health gain [26]. However, studies are lacking that demonstrate conclusive evidence that an early diagnosis is associated with decreased morbidity and mortality rates in a large cohort of agammaglobulinemia patients. Severe B-cell deficiencies can be detected by measuring kappa-deleting recombination excision circles (KRECs) in dried blood spots (DBS). Similar to T-cells, B-cells undergo V(D)J recombination to develop unique B-cell antigen receptors which also yield an excision circle: the KREC, serving as an indirect marker for the presence of B-cells [7,29]. KRECs can be measured simultaneously with TRECs in a multiplex qPCR-based assay allowing a swift implementation of KREC detection in the NBS laboratory at relatively low cost [30]. Detection of XLA and other B-cell deficiencies by KREC quantification in DBS has already been proven to be successful by several NBS pilot studies [31,32,33,34,35]. The reason why countries are not moving forward with NBS for B-cell deficiencies while a suitable test is available is probably due to the relatively high referral rate associated with KREC screening (0.08–0.1% [31] and our own data not published) and the lack of conclusive evidence of substantive health gain by early diagnosis of agammaglobulinemia. The Dutch Health Council proposed XLA as a potentially suitable candidate for NBS in 2015 but considered detailed identification of the exact characteristics of the test in routine neonatal screening a requirement [36]. Even though the referral rate in KREC screening would be depending on the chosen cut-off value, there is a need to evaluate second-tier options including epigenetic immune cell counting and next-generation sequencing (NGS) after KREC analysis. Finally, a retrospective multi-center study comparing clinical outcomes and quality of life of patients with an early and late diagnosis of XLA and other B-cell deficiencies will help NBS programs to move forward towards universal NBS for agammaglobulinemia resulting in health gain for these patients worldwide.

## 3. Epigenetic Immune Cell Counting: A New Player in the Field

Not all IEI and immune dysregulation disorders can be detected by absent TRECs or KRECs. With epigenetic immune cell counting, quantitative defects of immune cell populations such as T-cells, B-cells, regulatory T-cells (Tregs), and neutrophils could offer early detection of several IEI shortly after birth [37]. Epigenetic immune cell counting is a technique based on the amplification of cell-specific demethylated genomic regions with qPCR allowing measurement of relative cell counts in DBS as depicted in Figure 1 [37].

The absence of TRECs is a highly sensitive marker for SCID and epigenetic immune cell counting could only match this sensitivity if naïve T-cells or recent thymic emigrants (RTEs) could be quantified by epigenetic qPCR, i.e., to detect SCID cases with maternal engraftment [2,38]. Combined immunodeficiencies such as ZAP-70 deficiency or major histocompatibility complex (MHC) class I and II gene expression deficiency cannot be detected with the TREC assay as T-cell development is intact beyond the point of T-cell receptor (TCR) gene recombination [34]. MHC class I deficiency is characterized by a decreased surface expression of HLA class I molecules leading to decreased numbers of circulating CD8+ αβ T-cells, chronic infections in the respiratory tract, and skin granulomatous lesions. Prevention and treatment of bronchial infections are the main therapeutic strategies for these patients [39]. MHC class II deficiency leads to an impaired antigen presentation by antigen-presenting cells and incomplete maturation of CD4+ T-cells. Early diagnosis of MHC class II deficiency is important to enabling HSCT before irreversible organ damage secondary to recurrent infections has occurred [40]. As these CIDs usually present with low CD4+ or CD8+ T-cells, some patients could be identified with epigenetic immune cell counting shortly after birth. With epigenetic immune cell counting, NBS for XLA and other B-cell deficiencies based on quantification of relative B-cell counts could have a higher positive predictive value in comparison to KREC detection in DBS. Relative B-cell counts are after all a more direct marker for absolute B-cell counts than KRECs. By determining relative numbers of FOXP3+ Tregs, immune dysregulation disorders characterized by low Tregs could be identified in the neonatal phase. Monogenic autoimmune disorders caused by inborn errors in Tregs can have variable clinical manifestations, ranging from early-onset severe autoimmunity to late-onset or atypical symptoms [41]. Patients with an early-onset, severe phenotype require immediate therapy including immunosuppression followed by HSCT [42]. However, Treg numbers and function can be impaired by various underlying causes and NBS based on detection of FOXP3+ Tregs might be of limited value. In addition, quantification of relative Treg cell counts might not be an option for patients who express fairly normal amounts of mutated FOXP3 protein which is the case in some immune dysregulation polyendocrinopathy enteropathy X-linked (IPEX) syndrome patients [43]. Epigenetic immune cell counting did reveal increased levels of demethylation in the FoxP3 gene locus in symptomatic IPEX patients, potentially serving as a diagnostic aid [37]. With the relative quantification of neutrophils in DBS, severe congenital neutropenia (SCN) and other conditions associated with severe neutropenia at birth could be identified via NBS. Patients with SCN are characterized by impaired maturation of neutrophil granulocytes leading to recurrent, life-threatening infections and predisposition to myelodysplastic syndromes (MDS) or acute myeloid leukemia (AML) [44]. Daily subcutaneous G-CSF administration will lead to a reduction in infections, drastically improving quality of life. HSCT can serve as a curative treatment option for SCN patients who are nonresponsive to G-CSF therapy, patients who develop MDS or AML and patients with mutations in genes predisposing for malignant transformation (e.g., *CSF3R* or *RUNX1*) [45,46]. Prevention of infections would be the main purpose of early identification of SCN patients’ malignant transformation would occur at a later stage. A major challenge to overcome NBS for SCN would be the high number of secondary findings as neutropenia is frequently observed in neonates with maternal pre-eclampsia, sepsis, twin-twin transfusion, alloimmunization, and hemolytic disease being the most common causes [47]. Second-tier testing with NGS might be required to overcome this exceeding number of referrals [48]. In addition to NGS, repeating neutrophil measurements with epigenetic immune cell counting in second heel prick cards after one to two weeks could also reduce the high number of referrals as many of the above-mentioned causes of neonatal neutropenia will resolve shortly after birth. Figure 2 shows some additional immune types that can be identified with epigenetic immune cell counting and their corresponding quantitative defects or IEI. In addition to population-based screening, retrospectively applying epigenetic immune cell counting to NBS cards could allow the identification of neonatal prognostic markers for a range of disorders. The technique also facilitates diagnostics or monitoring in resource-poor regions, where logistics for appropriate cell counting is hampered as blood collection and measurement cannot be performed in close succession [37]. A pitfall of measuring relative cell counts in contrast to absolute cell counts as measured via flow cytometric immunophenotyping is that proportional cell numbers within the corresponding reference range might not accurately reflect the clinically relevant alterations in the patient. Patients could have very low numbers of total leukocytes with a normal percentage of T-cells concealing severe T-cell lymphopenia. Before epigenetic immune cell counting could be applied as a first-tier test in the screening laboratory, automating the protocol would be required to increase the throughput time and to enable an analysis of more samples with less hands-on time.

## 4. Newborn Screening for Interferonopathies

Another group of IEI potentially suitable for future NBS is type I interferonopathies. Type I interferonopathies encompass a spectrum of rare, genetic disorders that are characterized by autoinflammation and chronic type I interferon (IFN) production in the absence of a viral infection. In addition to elevated type I IFN levels, these disorders are characterized by calcifications in the central nervous system, leukoencephalopathy, severe developmental delay, and skin lesions. Because of the severity of these diseases, patients usually do not survive into adulthood [49,50]. Elevated type I IFN levels lead to an increase in ‘IFN-stimulated genes’ (ISGs). These ISGs can easily be monitored through quantitative PCR in peripheral blood. The results of a panel of six ISGs can be combined into an IFN score and this assay has been proposed as the ‘gold standard’ for the diagnosis of pediatric patients suffering from interferonopathies [51]. A recent multi-national study successfully showed that the IFN-score can be measured in DBS of newborns, allowing detection of type I interferonopathies shortly after birth [52]. Case reports with experimental treatments such as nucleoside reverse transcriptase inhibitors (NRTIs), IFN and IFN receptor blocking antibodies, and JAK1 inhibitors have suggested that early treatment may inhibit or delay developmental decline and disease progression [53]. More evidence will need to be collected on the effectiveness of these experimental treatments as actionability is one of the major criteria when considering conditions for NBS programs. In addition, the specificity of the IFN score as an NBS test should be determined in the context of patients with a viral infection who can also present with an evident IFN score [50].

## 5. Protein-Based Newborn Screening for IEI

In addition to DNA-based techniques, protein-based methods can also serve as potentially suitable screening tests for some IEI. Protein profiling has been described as a technique to broaden NBS for IEI with screening for innate immunity defects [54,55]. Recently, an NBS test based on suspension bead arrays for protein profiling has been described to detect 22 disorders due to defects in the complement system or phagocytic function prior to the onset of clinical symptoms [56]. Accurate and early diagnosis of these patients is important as complement deficiencies and phagocytic disorders are associated with numerous immunological complications. Complement deficiencies give rise to a variable clinical phenotype including recurrent and persistent infections, hereditary angioedema, and autoimmune complications. Disorders of granulocyte number and function lead to delayed wound healing, severe infections, abscess formation, and inflammatory manifestations (e.g., colitis in chronic granulomatous disease) [3]. Early diagnosis of such disorders allows immediate clinical intervention and prevention of severe morbidity and mortality. Some phagocytic diseases might even qualify for HSCT or in the future, gene therapy. A proteomic screening approach using tandem mass spectrometry was additionally described to quantify signature peptides for BTK, WASP, and T-cell marker CD3ε to screen for XLA, Wiskott–Aldrich Syndrome (WAS), and SCID, respectively [57]. WAS is a rare, X-linked IEI characterized by recurrent infections, microthrombocytopenia, eczema, and an increased incidence of autoimmunity and malignancies [58]. Mutations in the *WAS* gene have various effects on the level of WASp correlating to the severity of the disease. The absence of functional WASp can lead to fatal outcomes if not diagnosed and treated early in life with HSCT [58]. The selected reaction monitoring (immuno-SRM) technology further enhanced the sensitivity of quantifying IEI specific peptides with tandem mass spectrometry [59]. Recently, the proteomic panel was expanded to eight signature peptide biomarkers to screen for five molecularly defined IEI including adenosine deaminase (ADA) deficiency, dedicator of cytokinesis 8 (DOCK8) deficiency, X-Linked chronic granulomatous disease (XL-CGD), WAS, and XLA [60]. These IEI are strong candidates for inclusion in NBS programs as these disorders have effective treatment options, are well studied with a good understanding of the clinical course and immuno-SRM is highly suitable as a high-throughput test in the NBS laboratory. A key benefit of protein-profiling is the notable number of IEI-associated proteins that can be examined in parallel using a limited amount of sample material.

## 6. Genomic-Based Newborn Screening for IEI

Even though TREC screening was the first high-throughput DNA technique in the screening laboratory, targeted DNA sequencing is already used as a tiered screening strategy for cystic fibrosis [61,62]. Targeted DNA sequencing has also been described as a potential method to identify infants with familial hemophagocytic lymphohistiocytosis (FHLH) due to homozygous UNC13D inversion mutations [63]. Patients with HLH present with life-threatening inflammatory responses secondary to impaired lymphocyte functions. Clinical manifestations can include fever, splenomegaly, cytopenia, hypertriglyceridemia and/or hypofibrinogenemia, hemophagocytosis, low or absent NK-cell activity, hyperferritinemia, and elevated levels of soluble IL-2 receptor [64]. Early diagnosis is crucial to prevent severe disease manifestations by timely initiation of first-line treatment, to determine the need for HSCT, and to reduce possible post-HSCT sequelae [65]. There are several genes associated with FHLH [3], therefore, in order to identify all variants in genes associated with FHLH, other DNA-based techniques such as NGS should be considered.

Recent technological advances in genomic medicine have led to the availability of rapid and inexpensive genomic sequencing techniques, including NGS, whole-exome sequencing (WES), and whole-genome sequencing (WGS). TREC/KREC screening is unable to detect many serious IEI and immune dysregulation disorders and sequencing could provide a potential method for screening a wider array of health conditions. The increased use of NGS, WES, and WGS in diagnostics raises the question of whether these sequencing techniques could be applied in a screening context. Genomic-based NBS may be especially applicable to the detection of IEI, as these represent a heterogeneous group of conditions with varying clinical phenotypes. In addition, many IEI are monogenic, some of which may be difficult to diagnose clinically, and most can benefit from early medical interventions [66,67]. Given the genetic and phenotypic heterogeneity of IEI, screening all of these diseases would require a range of different test modalities, which is unfeasible from a logistic or economic perspective in the context of NBS. Applying genomic sequencing techniques in NBS would allow parallel testing, using one platform to detect many clinically actionable diseases [14]. The future role of genomic technology in NBS for IEI has previously been discussed extensively, therefore, this review will summarize the discussion points [14,68,69].

There are programs that have already successfully adopted NGS in their screening programs for SCID, primarily as a second-tier test after TREC analysis [70,71]. NGS with targeted gene panels on DBS will facilitate and accelerate final molecular diagnoses of affected newborns while providing useful information for management and follow-up. Previously, the time from sample collection to NGS results took weeks to months, but targeted NGS has a rapid turn-around time (results within 2–3 working days) [70]. Additionally, a higher TREC cut-off value in combination with NGS allows the detection of atypical and leaky SCID with potentially higher TREC values, but a clear HSCT indication based on immunophenotyping. On the other hand, NGS is associated with relatively high analyses and equipment costs and a cost-effectiveness analysis including efficiency gains and improved management could help NBS policymakers when discussing implementation of NGS [72]. The successful implementation of NGS in NBS as a second tier has opened the discussion for expansion of NBS for IEI by using sequencing techniques as a first tier [69]. NGS is very adaptable and could serve as a first-tier test to screen for monogenetic disease, however, exome-based targeted NGS will not be able to identify IEI with variants in genes not included in the gene panels or IEI with structural variants or intronic variants. Experts prefer WGS approaches as they are able to simultaneously sequence both intronic and exonic regions [14]. Although a proof-of-concept study for a WGS-based approach in screening for IEI has already been published, WGS poses significant challenges in the context of NBS [67]. Major concerns include the interpretation and managements of large amounts of genetic data and ethical implications of incidental findings and carrier status for patients and other family members. In addition, genome-scale sequencing would require modification of current informed consent procedures [69,73]. Pathogenicity interpretation and assessing the potentially deleterious effects of novel variants remains challenging. Even with automated methodology allowing high-throughput analysis of large amounts of genomic data, a manual review would still be required to define benign and pathogenic variants [67]. This is a labor-intensive, costly process and this type of expertise is currently not present in the majority of NBS laboratories. In addition, there is a need to improve the accuracy and completeness of reference databases and new methods for pathogenicity prediction are necessary before genomic testing can be incorporated into NBS programs.

Policymakers, NBS practitioners, clinicians, and parents have also raised social concerns about the expansion of NBS with WGS regarding privacy, trust, and desire for control over one’s own and one’s child’s genomic information [74]. Parents seem to have an overall optimistic and enthusiastic orientation towards genomic advances in NBS, but they expressed concerns about privacy and control over test results [74,75,76]. Genetic profiling and potential genetic discrimination are important aspects that would need to be addressed [77]. Due to limited trust in the medical system and the NBS programs, parents would desire more clarity over the data produced with genomic technologies. At this point, NBS stakeholders are uncertain how to manage unintended findings unrelated to actionable disorders and how to establish criteria for the evaluation and incorporation of new disorders. NBS programs and pediatricians will be responsible for the follow up of a greater number of conditions, as well as implementing an informed consent process and management of the genomic data produced by the test [74]. All technical challenges—as well as ethical, policy, and clinical practice issues—must be taken into consideration before adopting genomic technologies in population-based screening programs.

The adoption of TREC analysis and qPCR technology by NBS laboratories will enable further expansion of genomic techniques in NBS laboratories. However, several limitations, challenges, and important considerations must be addressed prior to routine implementation of genomic technologies in NBS programs. Many of the questions posed above remain unanswered and must be further evaluated and clarified in prospective studies assessing the entire screening process including ethical, legal, and social implications [69]. Screening without including any phenotypical markers as a first-tier option remains challenging due to the rarity of IEI, missing links between gene defects and disease mechanisms, and the inability to distinguish underlying pathogenic variants from the high number of genomic variations [52]. With genome-wide association studies, relations between phenotype traits and genotype in IEI might be unraveled. It remains to be seen whether the genomic technology will be used as a primary ‘standalone’ screening approach, or as an addition to current screening methodologies [14].

## 7. Future Newborn Screening for SCID

In the near future, SCID will be implemented by an increasing number of NBS programs worldwide. However, as the TREC assay is a relatively expensive technique, implementation of this method in screening laboratories might be challenging for countries with less resources. NBS for SCID is particularly important in some of these countries; for example in Middle Eastern countries where the incidence of IEI is expected to be 20 times higher than in North America or Europe due to the relatively high incidence of consanguinity [78]. Development of new, low-cost technologies for testing newborns for a broad range of conditions is key in this process, while commercial initiatives for innovative pricing of reagents and equipment can be of aid as well. Experts from various disciplines should contribute their time for training and sharing expertise on an international level [79]. The importance of screening programs cannot be outweighed, however, due to the lack of resources, educational programs, and public awareness campaigns might be a more feasible option in the direct future. In the absence of NBS, clinicians should be aware of the early manifestations of SCID to enable an early diagnosis and timely intervention [80]. The close partnership of NBS programs, policymakers, immunologists, and HSCT specialists and sharing of experiences internationally could help to improve outcomes for SCID patients on a global level.

While some NBS programs are still awaiting governmental decisions with regard to SCID screening, NBS programs that have already implemented SCID should continue to improve their current practice. With TREC screening, new cut-off values, adjusted screening algorithms, and inclusion of second-tier tests should be considered to increase positive predictive value and to reduce the number of false-positive referrals [81]. In addition, follow-up after an abnormal screening result will need to be further evaluated and compared across screening programs as follow-up diagnostics are not straightforward and might not always lead to a final diagnosis. Moreover, follow-up protocols need to be further optimized as previous studies have shown that a relatively large part of SCID patients identified with NBS still developed infections prior to HSCT [6]. The time required to obtain TREC results, to refer a newborn to a pediatric-immunologist, and to obtain results of confirmatory testing should be reduced to prevent significant delay in initiating protective measures. Best practice for isolation and antimicrobial prophylaxis to minimize infection exposure pre HSCT should be harmonized across centers [82].

In the coming years, more NBS programs might shift from SCID as the primary target towards screening for actionable T-cell lymphopenia with low TRECs. NBS with TREC testing correlates with having recently formed T-cells in peripheral blood; therefore, one could argue that in TREC-based screening primary targets should include all serious, actionable T-cell deficiencies that are associated with low TRECs at birth. Parents believe that the term actionable includes conditions (1) where early interventions lead to health gain for the newborn, (2) where early diagnosis avoids the lengthy diagnostic odyssey and (3) where parents will have reproductive options during subsequent pregnancies [83]. For many health care providers, the definition of actionability in NBS is more limited to the management of the individual affected with the condition. The term actionable indicates that an urgent (early) intervention is required by a specialist and that the intervention results in a demonstrable improvement in outcome. Cases of significant T-cell lymphopenia that might benefit from antibiotic prophylaxis and protective isolation should be deemed actionable. The term actionable is more suitable than the term treatable, as withholding live-attenuated vaccines is an important early intervention leading to improved outcomes, given that vaccine-strain organisms such as BCG can cause serious infections in individuals with T-cell defects [84,85].

In the next five years, many NBS programs will have implemented a multiplex PCR, measuring TRECs simultaneously with KRECs and *SMN1* introducing NBS for XLA and spinal muscular atrophy (SMA) [86]. The addition of KRECs will also be of value to SCID screening as it may assist in distinguishing B−/B+ phenotypes in SCID patients, therefore aiding in the diagnostic process. Some leaky or delayed-onset SCID patients, in particular T-B- SCID patients with hypomorphic mutations in DNA repair or cellular metabolism, might not be detected with TREC quantification [87]. The increase in toxic metabolites can well be tolerated to a certain degree by dividing T-cells, whereas B-cells seem to be more vulnerable for genomic stress, for example, in patients with delayed-onset ADA-SCID. For these patients, SCID screening will be further extended to tandem mass spectrometry measuring adenosine and deoxyadenosine for ADA deficient patients or purine nucleosides and 2′-deoxy-nucleosides for PNP deficient patients. Previous studies have shown that screening with tandem mass spectrometry was able to identify these infants at a low cost [88,89,90,91].

After these developments, epigenetic immune cell counting will hopefully be optimized as a high-throughput test for the NBS program enabling NBS for a range of IEI. We might be a long way from first-tier WGS-based screening in newborns, but a greater number of countries will include NGS in their NBS programs as a second-tier test in the next decade. Decades from now, NBS for IEI will enter the genomic era. Genome-wide association studies may have identified an exceeding number of associations between variants and phenotypes, explaining the contribution of common variants to variable penetrance and phenotypic complexity in IEI [92]. Reference databases will be more complete and pathogenicity prediction programs will demonstrate improved accuracy. Dried blood might no longer be the preferred material for neonatal screening as DNA can be obtained via less invasive techniques such as saliva or oral mucosa. In this era, non-actionable diseases might be included in the NBS program to avoid long diagnostic odysseys. In addition, NBS for early-onset diseases might have been complemented with conditions presenting in adulthood conflicting with the ‘child’s right to an open future’. Even risk scores of potentially developing a certain disease at some stage in life might be reported early in life. The future of NBS holds many uncertainties, but one thing is sure, with all these technological advances, exciting times are waiting for population-based screening programs.

## 8. Conclusions

In conclusion, NBS programs continue to expand with new conditions due to innovations in both test methods and treatment options. NBS for SCID based on TREC-detection was the first high-throughput DNA technique implemented in screening laboratories. In addition to SCID, there are many other IEI that could benefit from early diagnosis and intervention by preventing severe infections, immune dysregulation, and autoimmunity if a suitable NBS test was available. In the next years, the role of KREC analysis, epigenetic immune cell counting, IFN signatures, protein profiling, and genomic technologies for NBS for IEI will have to be further evaluated in the context of the entire screening process. In addition, other screening criteria and principles including ethical, social, and legal implications, logistics and costs will have to be carefully examined before different IEI can be considered as suitable candidates for inclusion in NBS programs.

## Figures and Tables

**Figure 1 IJNS-07-00074-f001:**
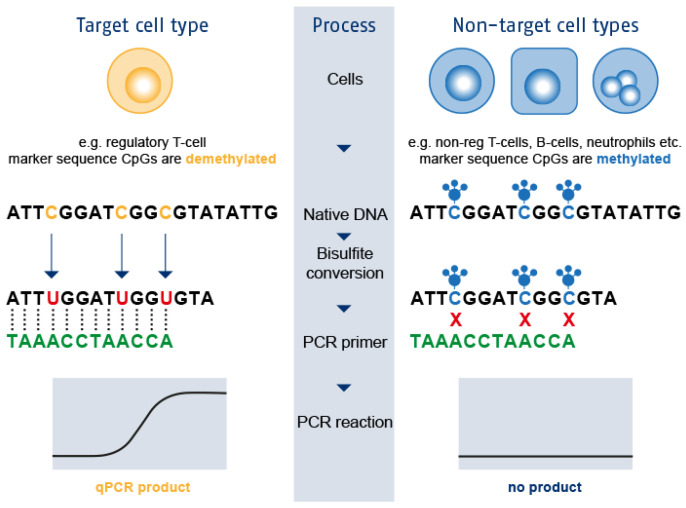
Epigenetic immune cell counting. Unique cell type-specific DNA methylation markers were identified. After bisulfite conversion of the genomic DNA, unmethylated CpG dinucleotides are converted and amplified to TpGs, whereas methylated CpGs remain unaltered. Bisulfite conversion translates epigenetic markers into sequence information, allowing immune cell quantification with qPCR [37]. Figure from Epimune GmbH, Berlin, Germany.

**Figure 2 IJNS-07-00074-f002:**
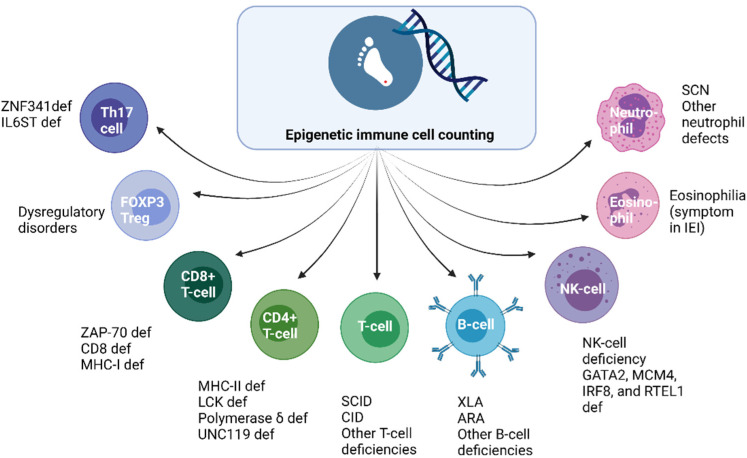
Different types of immune cells that can be identified with epigenetic immune cell counting and examples of corresponding quantitative defects or IEI. Th17—T-helper 17, def—deficiency, IL6ST—IL6 signal transducer, ZNF341—zinc finger protein 341, FOXP3—forkhead box P3, Treg—regulatory T-cell, ZAP-70—zeta-chain-associated protein kinase 70, MHC—major histocompatibility complex, SCID—severe combined immunodeficiency, CID—combined immunodeficiency, XLA—X-linked agammaglobulinemia, ARA—autosomal recessive agammaglobulinemia, NK-cell—natural killer cell, MCM4—minichromosome maintenance complex component 4, IRF8—interferon regulatory factor 8, RTEL1—regulator of telomere elongation helicase 1, SCN—severe congenital neutropenia, IEI—inborn error of immunity.

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
