# Peer review of "Future Perspectives of Newborn Screening for Inborn Errors of Immunity"

_2409-515X, 2021, doi:10.3390/ijns7040074_

Round 1

Reviewer 1 Report

This article comes from a team of authors with long-term experience with congenital immune disorders and, at the same time, with practical experience in the field of screening these diseases. This article presents a very instructive outlook for the future. It gives an overview of new technical possibilities that are already in place and proven is smaller, laready published studies. Such an overview is very important for specialists in the field, but also for policy makers and also for patients and their organizations as it outlines the possible development of the whole area of ​​screening for inborn errors of immunity in the future.

There are pracitcally no comments to be raised about the article, perhaps only a wide range of candidate diseases suitable for screening and equally large breadth of potential novel technologies. A graphic abstract would help in this, outlining the possibilities of practical application of the individual methods discussed in the article in real life. Many of the suggested methodologies overlap in the detection of individual diseases, and it is not possible or suitable to use the whole array of methods simultaneously.

The article mentions only marginally issues of a legislative nature, issues of personal data protection and hardly includes information on feasibility and the economy. However, due to the futuristic tuning of the article, these questions are not urgent yet.

The article deserves a publication for an excellent overview of current possibilities, but especially for the momentum that gives the field of screening for congenital immune disorders in the future.

Author Response

Please see the attacment.

Reviewer 2 Report

The authors review the current progress in newborn screening for inborn errors of immunity. The review includes more matured programs like SCID, under development like X-linked agammaglobulinemia (XLA), and the coming technologies like Epigenetic immune cell counting. This is a very nice prospective review of this field. There are a few comments that may benefit readers in the clinical fields.

  1. Confirmatory diagnosis for SCID is not straight forward, ranging from classic SCID to asymptomatic lymphopenia. The authors can address this issue in a more prominent way.
  2. Regarding the screening for XLA, the authors mention the difficulties including high referral. It may be helpful if the authors can show some real data.
  3. For Epigenetic immune cell counting, is there an algorithm that we can starting from identifying a few cell types rather than profiling all the cell types at a time.
